# DATA- AND HARDWARE-AWARE ENTANGLEMENT SELECTION FOR QUANTUM FEATURE MAPS IN HYBRID QUANTUM NEURAL NETWORKS

## ABSTRACT

Embedding classical data into a quantum feature space is a critical step for Hybrid Quantum Neural Networks (HQNNs). While entanglement in this feature map layer can enhance expressivity, heuristic choices often degrade trainability and waste the limited multiple-qubit gate budget. We reframe the choice of encoding-layer entanglement as a multi-objective combinatorial optimization that jointly promotes data-driven trainability and hardware-aware noise robustness. Our framework searches over sparse entanglement patterns by maximizing a novel data-utility term, balanced against a realistic hardware cost derived from device topology and calibrated two-qubit fidelities on IBM Quantum systems. The data-utility term pairs qubits based on two complementary geometric criteria: (i) high intrinsic dependency and (ii) low Hilbert–Schmidt Distance (HSD), a combination critical for accelerating gradient-based optimization early in HQNNs training. We solve this with a bi-level optimization scheme: the outer loop searches over discrete entanglement structures, evaluating each candidate's potential based on the initial loss reduction from a short inner-loop training. Once the optimal structure is identified, the downstream ansatz is trained to full convergence. Our empirical results demonstrate that suggested feature maps not only achieve superior classification performance on synthetic and real-world benchmarks but also demonstrate enhanced robustness under realistic noise models, all while maintaining a lower gate budget than heuristic designs. Our work establishes a principled, automated method for creating quantum feature maps that are simultaneously data-aware, hardware-efficient, and highly trainable.

## 1 INTRODUCTION

Recent advances in quantum hardware are driving innovation across many fields, with Quantum Machine Learning (QML) holding the promise of revolutionizing computation by exploiting phenomena like superposition and entanglement Liu (2023); Gujju et al. (2024); Abbas et al. (2024); Mazzola (2024); Ullah & Garcia-Zapirain (2024). Quantum Neural Networks (QNNs), including hybrid approaches, have emerged as a leading framework, demonstrating potential advantages in handling complex data correlations Bai & Hu (2024); Oliveira Santos et al. (2024); Shi et al. (2025); Huang et al. (2021); Gil-Fuster et al. (2024); Hafeez et al. (2024); Roh et al. (2024); Shi et al. (2024). However, a central challenge remains in understanding how to best harness these quantum properties.

Entanglement, a key quantum resource representing interactions between qubits, is crucial for QNNs but presents a significant challenge Horodecki et al. (2009). While essential for capturing complex feature correlations, naive or excessive entanglement can degrade performance by inducing barren plateaus and exacerbating the effects of hardware noise Ballarin et al. (2023); McClean et al. (2018). Furthermore,

two-qubit entangling gates are costly operations on near-term devices, being more error-prone and often requiring computationally intensive SWAP gates to match hardware topology Sweke et al. (2021); Baiguera et al. (2024). This highlights a critical need for principled entanglement strategies that balance expressibility with trainability and hardware constraints.

While much QNN research has focused on the parameterized ansatz, a growing body of work emphasizes that the initial data encoding step is equally critical for trainability Nagarajan et al. (2021); Paler et al. (2023); Fösel et al. (2021); Li et al. (2024); Zhang et al. (2024); Liao & Zhan (2022); Huang & Rebentrost (2023); Wierichs et al. (2022). The barren plateau phenomenon is fundamentally linked to the geometry of the quantum state space, which is quantitatively described by the Quantum Fisher Information (QFI). A well-designed feature map can favorably sculpt this geometric landscape, enhancing class separability and trainability, thereby reducing the burden on subsequent variational layers Lloyd et al. (2020); Hur et al. (2024). However, most studies treat data encoding and entanglement composition independently, despite their inherent interdependence. The structure of the encoded state dictates how entanglement propagates, and the entanglement profile, in turn, affects the representational power of the features.

To address this research gap, we introduce a framework that formulates entanglement selection as a formal multi-objective optimization problem. Our approach jointly considers three competing objectives: 1) a data-driven utility function to maximize trainability, 2) a hardware-aware cost accounting for device connectivity and gate errors, and 3) an efficiency regularizer promoting shallow circuits. At the core of our data utility is the Hilbert-Schmidt Distance (HSD), a metric used to select qubit pairs whose entanglement is theoretically linked to increasing the QFI and accelerating gradient-based learning. The outcome is a unified framework for automatically discovering optimized quantum feature maps under realistic hardware constraints.

Our main contributions are as follows:

- A novel framework for entanglement selection formulated as a multi-objective optimization problem. We reframe the heuristic art of entanglement design into a principled search that simultaneously optimizes for data-driven trainability, hardware-aware costs (topology, gate fidelity), and circuit efficiency.

- The establishment of a theoretical link between data geometry and QNN trainability. We theoretically and empirically demonstrate that pairing qubits with low Hilbert-Schmidt Distance (HSD) is a concrete mechanism for accelerating gradient-based optimization by directly influencing the Quantum Fisher Information (QFI).

- A practical and automated bi-level search algorithm for discovering robust QNN architectures. Our algorithm efficiently navigates the vast combinatorial space of entanglement structures, yielding circuits that not only achieve superior performance but also exhibit enhanced robustness to hardware noise.

## 2 RELATED WORK

Our work intersects with Quantum Architecture Search (QAS), data-driven feature map design, and hardware-aware circuit construction.

**Quantum Architecture Search (QAS)**   QAS frameworks aim to automate the design of variational circuits by searching for optimal structures, often optimizing for task performance while considering hardware costs like depth or SWAP penalties Zhang et al. (2022); Chen et al. (2024). Related pruning techniques use sensitivity or QFI-based metrics to sparsify circuits and mitigate barren plateaus Liu et al. (2024); Ohno (2024). While these methods typically optimize the entire ansatz holistically, our work focuses specifically on the foundational problem of selecting the initial entanglement structure within the data encoding layer itself.

**Data-Driven and Hardware-Efficient Ansatz Design** A significant body of work aims to tailor circuits to specific data or hardware. Data-driven methods engineer the feature map to reflect data structure. This includes techniques like quantum metric learning, which trains parameters to maximize class separability using metrics like HSD Hubregtsen et al. (2022); Gentinetta & Sutter (2023), and data re-uploading, which enhances model expressivity by repeatedly encoding inputs throughout the circuit Pérez-Salinas et al. (2020). More recent approaches such as Adaptive Pruning (ATP) analyze the input data to prune non-essential features before encoding, thereby reducing the required quantum resources and entanglement Afane et al. (2025). Separately, hardware-efficient ansätze are designed to respect device constraints, such as native gates and qubit connectivity, to minimize noise Kandala et al. (2017).

**QFI and Quantum Gradients** The QFI provides a fundamental upper bound on gradient magnitudes, establishing it as a direct measure of a circuit's learning potential Stokes et al. (2020). It also informs advanced optimization methods like the quantum natural gradient (QNG), which leverages the full QFI matrix to align the optimization path with the underlying geometry of the quantum state space.

## 3 SEARCHING THE OPTIMAL DATA ENCODING ENTANGLEMENT SELECTION WITH HARDWARE AND DATA AWARENESS

In this section, we present our framework for hardware-aware and data-driven entanglement structure search. We begin by defining the multi-objective function that guides our search, which holistically evaluates an entanglement structure based on data-driven metrics, hardware constraints, and classification loss functions. We then detail the bi-level combinatorial optimization algorithm designed to navigate the vast, discrete search space of possible entanglement topologies.

### 3.1 OVERVIEW: ENTANGLEMENT AS A MULTI-OBJECTIVE OPTIMIZATION PROBLEM

We move beyond heuristic approaches by formulating the selection of an optimal entanglement structure, denoted as a set of qubit pairs $M^*$, as a formal optimization problem. The goal is to find the structure that maximizes a comprehensive objective function $J(M, \theta^*)$, where $\theta^*$ represents the trained parameters of the Quantum Neural Network (QNN) for a given structure M. The objective function is composed of several, often competing, terms:

$$M^* = \arg \max_M J(M, \theta^*) = \mathcal{U}_{\text{data}}(M) - \alpha \mathcal{C}_{\text{hardware}}(M) - \beta \mathcal{R}_{\text{eff}}(M) \tag{1}$$

where $\alpha, \beta$ are hyperparameters that balance the trade-offs between the different objectives.

To ensure a fair and stable optimization across these potentially different-scaled objectives, all utility and cost terms are normalized before being combined. Specifically, we employ a rank-based scaling that maps the raw scores for all possible entanglement pairs onto a common [0, 1] interval, making the optimization robust to outliers and simplifying the tuning of the hyperparameters $\alpha$ and $\beta$.

### 3.2 DATA-DRIVEN UTILITY FUNCTION ($\mathcal{U}_{\text{DATA}}$)

This term evaluates the intrinsic potential of an entanglement structure based on the statistical and geometrical properties of the input data. The design of this term is grounded in enhancing the *trainability* of the QNN, which is fundamentally linked to the Gradient Norm and Quantum Fisher Information (QFI). The calculation process is as follows.

**State Preparation and Purification** First, we map the classical data into a quantum state. We employ a feature-wise Angle Encoding strategy, using the Y-axis rotation gate ($R_y$) which generates quantum states

with purely real amplitudes. From this, we compute an ensemble-averaged density matrix for each feature by averaging its quantum representations over the entire dataset. For an individual feature $i$, the averaged density matrix $\bar{\rho}_i$ is calculated as:

$$\bar{\rho}_i = \frac{1}{|\mathcal{D}|} \sum_{\mathbf{x} \in \mathcal{D}} |\psi(x_i)\rangle\langle\psi(x_i)| \tag{2}$$

A similar process is used to compute the two-qubit averaged density matrix $\bar{\rho}_{ij}$. To distill the core structural information from the noise of this classical averaging, we then *purify* these mixed states by projecting them onto their dominant eigenspace, yielding representative pure states $\tilde{\rho}_i$ and $\tilde{\rho}_{ij}$. (Detailed in Appendix A) This step allows our subsequent metrics to probe the quantum correlations more directly.

**Data-Driven Metrics**    Using these prepared states, we define two complementary metrics:

- **Hilbert-Schmidt Distance (HSD):** This metric quantifies the dissimilarity between the average states of two features. A low HSD indicates that the features map to similar regions in the quantum state space, which we found to be advantageous for achieving a high QFI and accelerating the initial increase of the gradient norm.

$$D_{\text{HS}}(\tilde{\rho}_i, \tilde{\rho}_j)^2 = \text{Tr}\left[(\tilde{\rho}_i - \tilde{\rho}_j)^2\right] \tag{3}$$

- **Quantum Correlation Metric ($\mathcal{I}_Q$):** This metric quantifies the dependency between a feature pair $(i, j)$ by measuring how far their joint representation is from a separable state. A high $\mathcal{I}_Q$ serves as the primary indicator for identifying feature pairs with a strong intrinsic relationship, making them candidates for entanglement. The mathematical derivation is discussed in Appendix B.

$$\mathcal{I}_Q(i, j) = D_{\text{HS}}(\tilde{\rho}_{ij}, \tilde{\rho}_i \otimes \tilde{\rho}_j)^2 \tag{4}$$

We adopt the Hilbert-Schmidt Distance (HSD) for its computational efficiency and straightforward estimability on near-term hardware via SWAP tests. While HSD is not Completely Positive and Trace-Preserving (CPTP)-monotone, our use of rank-1 projected states places our analysis in a regime where HSD maintains tighter connections to operational measures, mitigating theoretical concerns. (See Appendix C.)

**Combined Utility Function**    Our final utility function is designed to find a compromise that balances the structural benefit of high correlation (high $\mathcal{I}_Q$) with the need to maintain high trainability (low HSD):

$$\mathcal{U}_{\text{data}}(M) = \sum_{(i,j) \in M} \left( w_{\text{corr}} \mathcal{I}_Q(i, j) - w_H D_{\text{HS}}(\tilde{\rho}_i, \tilde{\rho}_j)^2 \right) \tag{5}$$

**Theory-consistent design**    The HSD term is used as a *geometric regularizer*: by favoring manifold-local pairings it controls sample-to-sample Hilbert–Schmidt distances within entangled neighborhoods and thus keeps the gradient field Lipschitz, preserving favorable (non–barren-plateau) scaling for local costs in early training (see Section 4.2 and Appendix E). While the largest *per-parameter* QFI gain $\Delta F_Q$ for a control rotation can arise from high-HSD pairings, repeatedly choosing such pairs across layers tends to enlarge effective light-cones and destabilize optimization; we therefore prioritize manifold-local pairings that maintain global trainability, even at the cost of a smaller immediate $\Delta F_Q$. Moreover, Appendix F shows that very large $\mathcal{I}_Q$ can suppress the attainable parameter–QFI for common post-entangler generators; accordingly, in Eq. equation 16 we treat $\mathcal{I}_Q$ as an informativeness signal but avoid unbounded maximization, combining it with the HSD regularizer and a gate-budget term. These choices complement the gradient–equator sensitivity link summarized in Appendix D.

### 3.3 HARDWARE-AWARE COST FUNCTIONS ($\mathcal{C}_{\text{HARDWARE}}, \mathcal{R}_{\text{EFF}}$)

**Hardware Cost ($\mathcal{C}_{\text{hardware}}$).**    This term quantifies the execution cost of the entanglement structure $M$ on a specific quantum device. We reference the **IBM Strasbourg** backend; however, in practice any backend for

the target real device can be used. We first construct a hardware graph $G_{\text{hw}}$ from the device's coupling map. For each pair $(i, j) \in M$, the cost is derived from the calibrated 2-qubit gate error rate $\epsilon_{uv}$ for each physical link $(u, v)$ on the hardware:

$$\mathcal{C}_{\text{hardware}}(M) = \sum_{(i,j) \in M} \left( \sum_{(u,v) \in \text{path}(i,j)} N_{\text{CNOT}}^{\text{SWAP}} \cdot \epsilon_{uv} \right) \tag{6}$$

where $\text{path}(i, j)$ is the shortest path between qubits $i$ and $j$ on $G_{\text{hw}}$. If $i$ and $j$ are not directly connected, the cost includes the accumulated error from the necessary SWAP gate sequence, where $N_{\text{CNOT}}^{\text{SWAP}} = 3$.

**Efficiency Regularizer ($\mathcal{R}_{\text{eff}}$).**  To promote sparse and resource-efficient circuits suitable for near-term quantum hardware, we introduce an efficiency regularizer, $\mathcal{R}_{\text{eff}}$. Instead of penalizing circuit depth directly, our regularizer controls the complexity of the entanglement structure $M$ using two distinct constraints:

1. **Total Budget Constraint:** A penalty is applied if the total number of entangling pairs, $|M|$, exceeds a predefined target budget, $B_{\text{target}}$. This directly controls the overall two-qubit gate count of the encoding layer.
2. **Per-Qubit Degree Constraint:** A penalty is applied to each qubit $i$ if its degree—the number of entangling gates it participates in—exceeds a specified threshold, $\tau_i$. This constraint prevents the formation of highly connected "hub" qubits, which can be particularly susceptible to crosstalk and accumulated error.

These constraints are implemented as penalty terms in the objective function. Their weights, represented by Lagrange multipliers ($\lambda_{\text{budget}}$, $\lambda_{\text{deg}}$), are dynamically updated during the bi-level search to guide the optimization towards structures that satisfy the desired sparsity and efficiency criteria.

This complexity measure is not a hard constraint but is incorporated into our main objective function as a Lagrangian penalty term. This approach creates a soft constraint that forces the bi-level optimization to find a trade-off between maximizing the data utility and performance versus minimizing the circuit depth. The strength of this penalty, and thus the importance of the trade-off, is controlled by a hyperparameter.

## 3.4 BI-LEVEL OPTIMIZATION FOR STRUCTURE SEARCH

Solving the main objective in equation 1 is challenging as it involves a complex interplay between a discrete, high-dimensional space of entanglement structures ($M$) and a continuous space of QNN parameters ($\theta$). To tackle this, we employ a **bi-level optimization** strategy, which decouples the problem into nested levels:

- **The Outer Level (Structure Search):** At this level, we search through the discrete space of candidate entanglement structures $M$ to find one that optimizes our main objective. This search is guided by the data-driven utility function $\mathcal{U}_{\text{data}}$ and the QNN's performance.
- **The Inner Level (Parameter Training):** For a given entanglement structure $M$ fixed by the outer level, we perform a standard continuous optimization over the QNN parameters $\theta$ to minimize the training loss, finding the optimal parameters $\theta^*$ for that specific structure.

A full inner-level training for every candidate structure would be computationally prohibitive. To make the search tractable, we introduce a **proxy objective** for efficiently evaluating candidate structures. After a full training run for the current best structure, we "warm-start" the evaluation of new candidates by training them for only a small number of epochs, starting from the previously optimized parameters. After training small steps, output of the loss function is compared with the previous inner loop. If loss improvement is observed, current structure is adopted, otherwise rejected. This provides a computationally cheap yet effective estimate of a candidate's potential and implicitly considers the classification performance. More detailed explanation is shown in Appendix G as algorithmic pseudo-code (Algorithm 1).

**Initialization with Data-Driven Heuristics.**    The search does not begin from a random point. Instead, we first compute the data utility $\mathcal{U}_{\text{data}}(i, j)$ for all potential edges. Based on this, we derive two key heuristics:

- **Degree Hints ($\tau$):** We calculate a 'node score' for each qubit by summing the utilities of all connected edges. A degree hint $\tau_i$ for each qubit $i$ is then computed proportionally to its score. This estimates the "ideal" number of connections for each qubit based on its importance inferred from the data.
- **Target Budget ($B$):** The overall target number of entanglement gates (the budget $B$) is determined by the sum of these degree hints: $B = \text{round}(\frac{1}{2}\sum_i \tau_i)$.

Using these heuristics, an initial structure $M_0$ is generated via a greedy matching algorithm that respects the degree hints. Lagrange multipliers for the budget and degree constraints ($\lambda_B, \{\lambda_i^{\text{deg}}\}$) are initialized to zero.

**Lagrangian Dual Update.**    To guide the search towards structures that satisfy our constraints, we employ a dual update mechanism. At the end of each outer round, we measure the constraint violations of the current structure $M_t$ (e.g., $|M_t| - B$ for the budget). The Lagrange multipliers are then updated via a **dual ascent**:

$$\lambda \leftarrow \max(0, \lambda + \eta \times (\text{constraint violation})) \tag{7}$$

This update increases the penalty for violated constraints in the next round. For instance, if the budget is exceeded, $\lambda_B$ increases, which in turn decreases the adjusted weights for adding new edges, thus steering the search back towards the budget. This creates a self-correcting feedback loop that balances the search for high-performing structures with the need to adhere to the target constraints.

## 4 EMPIRICAL EXPERIMENTS

To validate the effectiveness of our data-driven entanglement search methodology, we conducted a comprehensive set of experiments. To demonstrate its superiority and robustness, we benchmark our proposed Searched strategy against standard heuristics, primarily Linear and Random entanglement structures, as baselines (Sec. 4.1). This comparison is performed across multiple datasets and under three distinct data encoding environments Simple Angle Encoding, ATP, and Data Re-uploading to verify the generality of our method. Beyond benchmark performance, in Sec. 4.2 we explored more deeper into the training dynamics to provide evidence for the improved trainability and performance gain. To analyze the independent effect of our data utility function, we conduct an ablation study in Sec. 4.3. Noise injection environment experiment was also conducted to verifying the hardware efficiency, depicted in Sec. 4.4. Detailed experimental setups (i.e. hyper parameter, dataset description) are described in Appendix H.

### 4.1 BENCHMARK EVALUATION WITH IDEAL SIMULATOR (WITHOUT NOISE)

We evaluated our searched entanglement structures against **Linear** and **Random** baselines on a synthetic dataset with complex correlations and the real-world Heart UCI dataset. The comparison was performed across three contexts: a base angle encoding, an advanced ATP encoding, and a data re-uploading strategy. Table 1 summarizes the mean Area Under the Curve (AUC) and Accuracy on a held-out test set.

The results demonstrate the robustness of our approach. Our searched architectures consistently outperform the baselines across all settings. The advantage is most pronounced on the complex synthetic data, where our best-performing model (using data re-uploading) achieved an AUC of **0.9827**, far exceeding the Linear baseline's 0.6550. On the Heart dataset, our method also achieved the highest AUC (**0.9349**) and Accuracy.

### 4.2 TRAINING DYNAMICS

To validate the efficacy of our proposed data-driven entanglement search strategy, we conducted a comparative analysis of its training dynamics against several controlled heuristic strategies. These include strategies

Table 1: Benchmark results on Synthetic and Heart datasets under ideal simulation. Mean (and standard deviation) of AUC and Accuracy are reported. Best results in each column are in bold.

| | Synthetic Data | | Heart Data | |
|---|---|---|---|---|
| **Method** | **AUC** | **Accuracy** | **AUC** | **Accuracy** |
| Searched (ours) | 0.9756 (0.0091) | 0.9130 (0.0243) | 0.9279 (0.0169) | 0.8576 (0.0349) |
| Linear | 0.6550 (0.0295) | 0.6200 (0.0255) | 0.8754 (0.0104) | 0.7971 (0.0055) |
| Random | 0.8945 (0.0801) | 0.8150 (0.1011) | 0.9275 (0.0199) | 0.8557 (0.0415) |
| *ATP Encoding* | | | | |
| + Linear | 0.6430 (0.0452) | 0.6020 (0.0370) | 0.8857 (0.0075) | 0.8127 (0.0212) |
| + Searched (ours) | 0.9759 (0.0094) | 0.9130 (0.0243) | 0.9267 (0.0181) | 0.8572 (0.0374) |
| *Data Re-uploading* | | | | |
| + Linear | 0.8874 (0.0056) | 0.8150 (0.0110) | 0.8866 (0.0125) | 0.7980 (0.0231) |
| + Searched (ours) | **0.9827 (0.0078)** | **0.9250 (0.0225)** | **0.9349 (0.0102)** | **0.8673 (0.0138)** |

that solely prioritize low HSD, high correlation and a baseline model with a fixed, non-data-driven linear entanglement structure. The results of training dynamics were depicted in Figrue 1.

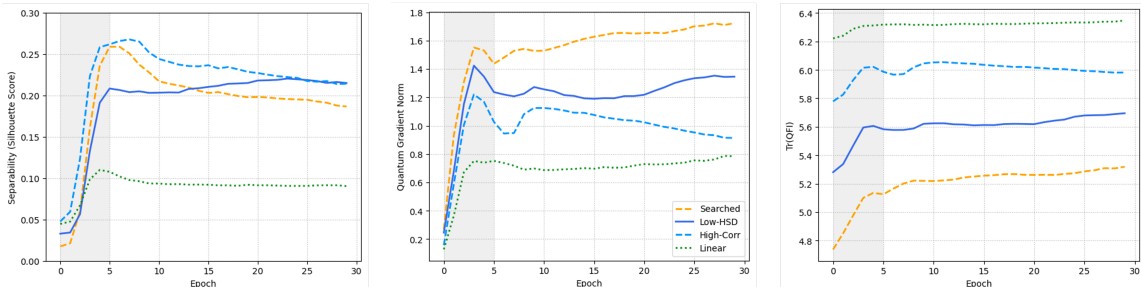

(a) Class separability in the quantum feature space, measured by the Silhouette Score.

(b) The magnitude of the quantum gradient norm, indicating the model's trainability.

(c) The trace of the Quantum Fisher Information (Tr(QFI)), a measure of the total learning capacity.

Figure 1: **Comparison of training dynamics for different entanglement strategies.** The shaded region highlights the initial training phase (epochs 0-5).

This experiment was conducted on an 8-qubit simulated dataset. The Low-HSD and High-HSD and correlation strategies entangled the seven pairs with the lowest and highest HSD (or correlation) values, respectively. For the Searched method, the entanglement counts were optimized to 4 which indicates fewer CNOT gate than each comparison.

The empirical results demonstrate that our proposed Searched strategy, along with the Low-HSD and High-Corr heuristics, achieves substantially better convergence, as evidenced by a higher class separability Figure 1a. Notably, this superior performance presents a clear contrast to the total learning capacity measured by Tr(QFI) Figure 1c, where these methods are not dominant. Instead, as detailed in Figure 1b the success is explained by more direct measures of trainability.

It supports our central hypothesis, theoretically detailed in Appendix D.1 that the alignment of the QFI with the optimization direction—captured by the Task-Aligned QFI and reflected in a robust Quantum Gradient Norm—is a more critical contributor to QNN performance than the total expressibility. We further discuss about Task-Aligned QFI in Appendix. I.

In essence, our data-driven search discovers a balanced architecture that harnesses the high class-separability promoted by data-feature correlations while retaining the efficient optimization landscape characteristic of low-HSD pairings, a mechanism mathematically formalized in Appendix E. Our proposed method exhibits rapidly increase in trajectory, especially during the initial training phase (epochs 0–5), where determine the overall learning direction.

### 4.3 ABLATION STUDY

To verify that both the intrinsic dependency ($\mathcal{I}_Q$) and the Hilbert-Schmidt Distance (HSD) components are essential to our utility function, we conducted an ablation study. We performed the entire bi-level search using two ablated utility functions: one guided solely by the HSD term (HSD Only) and another solely by the dependency term ($\mathcal{I}_Q$). We compare the performance of architectures found by these ablated searches against our Full Method on the synthetic dataset, with results summarized in Table 2.

Table 2: Ablation study results on the Synthetic dataset. The performance of the full utility function is compared against versions using only the correlation term ($\mathcal{I}_Q$ Only) or only the HSD term (HSD Only). The columns on the right show the change ($\Delta$) relative to the Full Method. The best overall performance is highlighted in bold.

| Utility Function | Encoding Strategy | AUC | Accuracy | AUC ($\Delta$) | Acc ($\Delta$) |
|---|---|---|---|---|---|
| Full Method ($\mathcal{I}_Q$ + HSD) | Searched (Simple Angle) | 0.9384 (0.0115) | 0.8722 (0.0249) | - | - |
| | ATP + Searched | 0.9365 (0.0122) | 0.8664 (0.0235) | - | - |
| | Re-uploading + Searched | 0.9361 (0.0101) | 0.8693 (0.0226) | - | - |
| HSD Only (w/o $\mathcal{I}_Q$) | Searched (Simple Angle) | 0.9372 (0.0119) | 0.8625 (0.0302) | ↓0.0012 | ↓0.0097 |
| | ATP + Searched | 0.9344 (0.0119) | 0.8605 (0.0288) | ↓0.0021 | ↓0.0059 |
| | **Re-uploading + Searched** | **0.9410 (0.0101)** | **0.8741 (0.0290)** | ↑0.0049 | ↑0.0048 |
| $\mathcal{I}_Q$ Only (w/o HSD) | Searched (Simple Angle) | 0.9223 (0.0177) | 0.8322 (0.0231) | ↓0.0161 | ↓0.0400 |
| | ATP + Searched | 0.9167 (0.0173) | 0.8410 (0.0243) | ↓0.0198 | ↓0.0254 |
| | Re-uploading + Searched | 0.9160 (0.0156) | 0.8349 (0.0215) | ↓0.0201 | ↓0.0344 |

The $\mathcal{I}_Q$ Only search consistently yields the lowest performance, confirming that maximizing correlation alone is an insufficient strategy. The HSD Only search is highly effective, achieving the single best result (AUC of 0.9410) when paired with data re-uploading. This underscores the critical role of the low-HSD criterion in establishing a trainable foundation for the model, as theorized in Appendix E.

For the remaining two encoding strategies, however, our Full Method achieved superior results. This suggests that while a pure HSD-driven search can discover highly expressive architectures, the Re-uploading strategy's reliance on deeper circuits can be a significant vulnerability in the presence of noise. This trade-off is explicitly demonstrated and discussed in the subsequent noise simulation experiments (Sec. 4.4).

### 4.4 NOISE INJECTION EXPERIMENT

We provide the noise simulation results with simulation dataset under 4-qubits and 7-qubits. The hardware-aware cost ($\mathcal{C}_{\text{hardware}}$) is calculated using the topology and calibrated gate fidelities from **ibmq Strasbourg**. To test for robustness, noisy simulations are performed using a noise model constructed from the pennylane `qubit.mixed` simulator. We simulated only 2-qubit gates error to eliminate other gate noise or read out error affect and purely the entanglement based error.

Table 3 highlights the superior performance and noise robustness of our Searched method. The identical performance between our Searched and ATP + Searched methods is due to the low-dimensional nature of the datasets used; the ATP pre-processing step did not identify any features to prune, resulting in the subsequent search being performed on the same feature set for both configurations. Notably, the Data Re-uploading

Table 3: Benchmark results on Synthetic datasets (4 and 7 qubits) under 2-qubit noisy simulator. Mean (and standard deviation) of AUC and Accuracy are reported. Best results in each column are in bold.

| | 4-qubits Synthetic Data | | 7-qubits Synthetic Data | |
|---|---|---|---|---|
| **Method** | **AUC** | **Accuracy** | **AUC** | **Accuracy** |
| **Searched (ours)** | **0.9615 (0.0093)** | **0.8880 (0.0173)** | **0.9789 (0.0041)** | **0.9220 (0.0077)** |
| Linear | 0.9162 (0.0125) | 0.8310 (0.0119) | 0.8145 (0.0211) | 0.7260 (0.0293) |
| *ATP Encoding* | | | | |
| + Linear | 0.9162 (0.0125) | 0.8310 (0.0119) | 0.8145 (0.0211) | 0.7260 (0.0293) |
| + Searched (ours) | **0.9615 (0.0093)** | **0.8880 (0.0173)** | **0.9789 (0.0041)** | **0.9220 (0.0077)** |
| *Data Re-uploading* | | | | |
| + Linear | 0.8959 (0.0051) | 0.7960 (0.0118) | 0.8490 (0.0188) | 0.7730 (0.0151) |
| + Searched (ours) | 0.9585 (0.0086) | 0.8880 (0.0209) | 0.9188 (0.0048) | 0.8250 (0.0089) |

strategy, which superior in ideal simulations, showed degraded performance. This is an expected outcome in a noisy environment, as its high expressivity is achieved through deeper circuits with more error-prone two-qubit gates.

This suggests that for near-term devices, raw expressive power must be carefully balanced against noise resilience. The clear superiority of our Searched method in the noisy simulation validates our dual-criteria approach, proving it is essential for discovering architectures that are both effective in theory and robust in practice.

## 5 CONCLUSION

In this work, we introduced and validated a data-driven strategy for co-designing QNN entanglement structures to be both powerful and practical for near-term hardware. Furthermore, it possesses the strength of being extensively applicable to various encoding schemes, ATP or Data Re-uploading, as shown in Section 4.1. Our method is centered on a multi-objective utility function that navigates the complex trade-offs inherent in variational algorithm design. This function synergistically combines performance-oriented terms—a quantum correlation measure ($\mathcal{I}_Q$) for informational potency and a low-HSD criterion acting as a crucial geometric regularizer to ensure global trainability—with pragmatic, hardware-aware constraints. These constraints, incorporated as Lagrangian penalties, explicitly penalize entanglement topologies that incur high hardware costs (e.g., large SWAP gate overheads or connections with high two-qubit error rates) and those that violate pre-defined circuit depth limits. Our theoretical and experimental results confirm that this integrated approach discovers fundamentally trainable and efficient entanglement structures, leading to superior outcomes characterized by high, stable gradient norms and the sculpting of a compact, task-aligned optimization landscape.

Looking forward, several avenues remain for future investigation. While direct validation on quantum hardware is a crucial next step, our framework's inherent consideration of hardware costs—supported by simulations including two-qubit gate errors (see Sec. 4.4)—suggests it is well-suited for robust performance on near-term devices. Future work should also address the scalability of the pre-processing step. The computational cost of calculating HSD and $\mathcal{I}_Q$ for very large datasets necessitates the exploration of more efficient optimization methods and effective approximation schemes. Furthermore, deploying this strategy on real devices will require moving beyond direct state vector access. Developing methods to estimate these geometric and informational metrics from measurement outcomes, for instance through state tomography or approximations from Bloch vectors, presents a vital and practical avenue for future research.

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

# A   APPENDIX A. STATE PURIFICATION PROCESS

This appendix details the mathematical procedure for the state purification (more formally, principal-eigenvector projection) used throughout this paper. This step is crucial for distilling a clear structural signal from the ensemble-averaged density matrices, which are inherently mixed states.

## A.1   MOTIVATION: FROM MIXED STATES TO REPRESENTATIVE PURE STATES

The ensemble-averaging process described in Section 3.2 produces a density matrix, $\bar{\rho}$, that represents the average quantum state for a given feature over the entire dataset $\mathcal{D}$. This averaging introduces classical statistical uncertainty, resulting in a mixed state. While this mixed state accurately reflects the overall distribution, our goal is to analyze the single, dominant geometric structure of the feature's quantum representation.

To achieve this, we project the mixed state $\bar{\rho}$ onto its "closest" pure state, $\tilde{\rho}$. This procedure effectively filters out the statistical "blur" from the classical averaging, allowing us to analyze the underlying quantum structure more directly. This projection is achieved by identifying the principal eigenvector of the averaged density matrix.

## A.2   MATHEMATICAL PROCEDURE

The projection from a mixed state $\bar{\rho}$ to its representative pure state $\tilde{\rho}$ follows a standard procedure from linear algebra and quantum information theory.

**Step 1: Spectral Decomposition.** Any density matrix $\bar{\rho}$ is a Hermitian, positive semi-definite matrix. According to the spectral theorem, it can be decomposed into its eigenvalues and eigenvectors:

$$\bar{\rho} = \sum_{k=1}^{d} \lambda_k |\phi_k\rangle\langle\phi_k| \tag{8}$$

where $d$ is the dimension of the Hilbert space, $\{\lambda_k\}$ are the real, non-negative eigenvalues, and $\{|\phi_k\rangle\}$ form an orthonormal set of eigenvectors. Since $\mathrm{Tr}(\bar{\rho}) = 1$, the eigenvalues sum to one, $\sum_k \lambda_k = 1$, and can be interpreted as the statistical weights of the pure state components $|\phi_k\rangle\langle\phi_k|$ in the mixture.

**Step 2: Identifying the Principal Eigenvector.** The dominant component in this mixture is the one corresponding to the largest eigenvalue. We define the largest eigenvalue as $\lambda_0 = \max_k\{\lambda_k\}$ and its corresponding eigenvector, $|\phi_0\rangle$, as the **principal eigenvector**. This vector represents the most probable pure state within the statistical ensemble described by $\bar{\rho}$.

**Step 3: Projection.** The final purified state, $\tilde{\rho}$, is defined as the projection onto this principal eigenvector:

$$\tilde{\rho} = |\phi_0\rangle\langle\phi_0| \tag{9}$$

This resulting pure state $\tilde{\rho}$ is the one that is closest to the original mixed state $\bar{\rho}$ as measured by fidelity, maximizing the quantity $\mathrm{Tr}(\sigma\bar{\rho})$ over all pure states $\sigma$.

### A.3 Application in Our Framework

In our work, this projection is applied to both the single-qubit and two-qubit ensemble-averaged density matrices:

$$\bar{\rho}_i \xrightarrow{\text{Projection}} \tilde{\rho}_i \tag{10}$$

$$\bar{\rho}_{ij} \xrightarrow{\text{Projection}} \tilde{\rho}_{ij} \tag{11}$$

The resulting purified states, $\tilde{\rho}_i$ and $\tilde{\rho}_{ij}$, are then used to calculate our data-driven metrics, such as the Quantum Correlation Metric $\mathcal{I}_Q$. This ensures that our metrics are comparing the core structural properties of the feature representations rather than the statistical noise from the averaging process.

## B Appendix B. Proof that $\mathcal{I}_Q$ Quantifies Qubit–Qubit Correlation

Let $\bar{\rho}_i \in \mathbb{C}^{2\times2}$ and $\bar{\rho}_{ij} \in \mathbb{C}^{4\times4}$ denote the empirical one- and two-qubit averaged density matrices, respectively, obtained from the feature encoding map. Define their pure projections

$$\tilde{\rho}_i := \Pi_{\text{pure}}(\bar{\rho}_i), \quad \tilde{\rho}_j := \Pi_{\text{pure}}(\bar{\rho}_j), \quad \tilde{\rho}_{ij} := \Pi_{\text{pure}}(\bar{\rho}_{ij}),$$

where $\Pi_{\text{pure}}$ denotes the projection onto the nearest pure state. We then introduce the correlation score

$$\mathcal{I}_Q(i,j) := \|\tilde{\rho}_{ij} - \tilde{\rho}_i \otimes \tilde{\rho}_j\|_{HS} = \sqrt{\mathrm{Tr}\left[(\tilde{\rho}_{ij} - \tilde{\rho}_i \otimes \tilde{\rho}_j)^2\right]}.$$

**Theorem 1.** $\mathcal{I}_Q$ *vanishes if and only if the two-qubit projected state factorizes as* $\tilde{\rho}_{ij} = \tilde{\rho}_i \otimes \tilde{\rho}_j$*. Hence,* $S_{ij}$ *is a faithful measure of correlation between qubits* $i$ *and* $j$*.*

*Proof.* (*If*). Suppose the data distribution is independent so that $p(x_i, x_j) = p(x_i)p(x_j)$. With a local (non-entangling) encoding map,

$$\bar{\rho}_{ij} = \frac{1}{n}\sum_s \rho_i^{(s)} \otimes \rho_j^{(s)} = \bar{\rho}_i \otimes \bar{\rho}_j.$$

Since $\Pi_{\text{pure}}$ acts locally on product states, we have

$$\widetilde{\rho}_{ij} = \Pi_{\text{pure}}(\bar{\rho}_{ij}) = \Pi_{\text{pure}}(\bar{\rho}_i) \otimes \Pi_{\text{pure}}(\bar{\rho}_j) = \widetilde{\rho}_i \otimes \widetilde{\rho}_j.$$

Thus $\mathcal{I}_Q(i,j) = 0$.

(*Only if*). Conversely, $\mathcal{I}_Q = 0$ implies that $\|\widetilde{\rho}_{ij} - \widetilde{\rho}_i \otimes \widetilde{\rho}_j\|_{HS} = 0$, which holds only if $\widetilde{\rho}_{ij} = \widetilde{\rho}_i \otimes \widetilde{\rho}_j$. Therefore the projected state factorizes, meaning no correlation is present in the pure-projected representation. $\square$

For pure states, $\mathcal{I}_Q(i,j)$ reduces to a fidelity-based form:

$$[\mathcal{I}_Q(i,j)]^2 = 2\,(1 - F), \quad F := \text{Tr}(\widetilde{\rho}_{ij}\,\widetilde{\rho}_i \otimes \widetilde{\rho}_j) \in [0,1].$$

*Proof.* Both $\widetilde{\rho}_{ij}$ and $\widetilde{\rho}_i \otimes \widetilde{\rho}_j$ are pure, so $\text{Tr}(\cdot^2) = 1$. Expanding,

$$[\mathcal{I}_Q(i,j)]^2 = \text{Tr}(\widetilde{\rho}_{ij}^2) + \text{Tr}\big((\widetilde{\rho}_i \otimes \widetilde{\rho}_j)^2\big) - 2\,\text{Tr}(\widetilde{\rho}_{ij}\,\widetilde{\rho}_i \otimes \widetilde{\rho}_j)\,.$$

Thus $[\mathcal{I}_Q(i,j)]^2 = 2 - 2F$. $\square$

**Interpretation.**

- $I_Q(i,j) = 0$ iff $\widetilde{\rho}_{ij}$ factorizes (no correlation).
- $I_Q(i,j) > 0$ iff there exists correlation, either classical (separable but not product) or quantum (entanglement).
- $I_Q(i,j)$ ranges between $0$ and $\sqrt{2}$, attaining the maximum when the two states are orthogonal.
- In the Bloch representation, if the marginals match, $[I_Q(i,j)]^2 = \frac{1}{4}\|\widetilde{T} - \widetilde{\vec{r}}\,\widetilde{\vec{s}}^{\,T}\|_F^2$, i.e. it measures the deviation of the correlation tensor from its factorized form.

**Remark.** Since Hilbert–Schmidt norm and fidelity are unitarily invariant, $I_Q(i,j)$ is invariant under local basis changes (e.g. RX vs. RY encoding). If a nonlocal entangler (e.g. CZ) is inserted, $I_Q(i,j)$ captures the induced correlations as well, which can increase its value even when the underlying features are independent.

## C APPENDIX C. CHOICE OF DISTANCE METRIC: HSD VS. TRACE DISTANCE VS. FIDELITY (AND CPTP MONOTONICITY)

**CPTP monotonicity.** A distance (or divergence) $D$ is called *CPTP-monotone* if it does not increase under any completely positive trace-preserving (CPTP) map $\Phi$, i.e.,

$$D\big(\Phi(\rho), \Phi(\sigma)\big) \leq D(\rho, \sigma) \quad \text{for all states } \rho, \sigma \text{ and channels } \Phi. \tag{12}$$

The trace distance $D_{\text{tr}}(\rho, \sigma) = \frac{1}{2}\|\rho - \sigma\|_1$ is contractive under CPTP maps, and the Uhlmann fidelity $F(\rho, \sigma) = \|\sqrt{\rho}\sqrt{\sigma}\|_1^2$ is monotonically non-decreasing under CPTP maps; accordingly, fidelity-induced metrics such as the Bures distance are CPTP-contractive. By contrast, the Hilbert–Schmidt distance $D_{\text{HS}}(\rho, \sigma) = \|\rho - \sigma\|_2$ is *not* CPTP-monotone in general, which limits its direct use as an operational distinguishability measure for arbitrary channels.

**Why HSD can still be appropriate here.**   Despite the above caveat, HSD retains attractive properties for our application. It is defined via a simple norm and admits efficient estimation on quantum hardware using SWAP-test circuits that provide overlaps and purities, enabling

$$D_{\mathrm{HS}}^2(\rho, \sigma) = \mathrm{Tr}(\rho^2) + \mathrm{Tr}(\sigma^2) - 2\,\mathrm{Tr}(\rho\sigma),$$

which explains its widespread use as a QNN cost Lloyd et al. (2020); Coles et al. (2019). On the other hand, Ozawa (2000) highlighted limitations of HS-based quantities for entanglement quantification—particularly for mixed, high-rank states—since non-contractivity can obscure physical meaning. In contrast, Coles et al. (2019) argued that for *low-rank* states (i.e., few non-zero eigenvalues relative to Hilbert-space dimension, close to pure) HSD can be tightly related to operational measures such as trace distance via rank-dependent bounds.

**Low-rank regime in our pipeline.**   In our framework (Sec. 3.2), state distances are evaluated immediately after a simple base encoding *without* entanglement, and we then form a rank-1 representative by the principal-eigenvector projection[1]. Although the ensemble average $\bar{\rho}$ can be high-rank, *our metrics are computed on the rank-1 representatives $\tilde{\rho}$.* This places us in the low-rank regime where HSD's relationship to operational distances is tighter, mitigating the main theoretical concern for HSD in general mixed-state settings Coles et al. (2019).

**Practical choice and safeguards.**   We therefore adopt HSD for its computational simplicity and near-term estimability, while acknowledging its lack of CPTP monotonicity in general. To safeguard trainability and relevance, we (i) optimize correlation contributions in the data-utility term ( equation 16) to avoid pathological regimes, and (ii) incorporate a short-horizon, task-aligned validation signal in the outer objective ( equation 1). This combination balances practicality with the theoretical caution raised in Ozawa (2000), leveraging the low-rank setting where HS-based distances admit stronger connections to operational measures Coles et al. (2019).

# D   APPENDIX D. THEORETICAL JUSTIFICATION FOR THE DATA-DRIVEN UTILITY FUNCTION

In this appendix, we provide the theoretical underpinnings for the data-driven utility function, $\mathcal{U}_{\mathrm{data}}$. We connect our metrics to the fundamental concepts of Quantum Fisher Information (QFI), the geometry of the Bloch sphere, and the geometry of the quantum data manifold.

## D.1   TRAINABILITY, GRADIENTS, AND THE ROLE OF THE BLOCH SPHERE EQUATOR

The trainability of a QNN is critically dependent on the magnitude of the gradients of the loss function. For a parameter $\theta_j$, this gradient is $\frac{\partial \mathcal{L}}{\partial \theta_j} = \frac{\partial \mathcal{L}}{\partial \langle Z_k \rangle} \frac{\partial \langle Z_k \rangle}{\partial \theta_j}$. The quantum gradient term, $\frac{\partial \langle Z_k \rangle}{\partial \theta_j}$, is bounded by the QFI and dictates the potential for efficient learning.

Consider a rotational gate $R_Y(\theta_j)$ acting on a state $|\psi_{\mathrm{in}}\rangle$. The gradient of the expectation value $\langle Z_j \rangle_{\mathrm{out}}$ with respect to $\theta_j$ is:

$$\frac{\partial \langle Z_j \rangle_{\mathrm{out}}}{\partial \theta_j} = -\langle Z_j \rangle_{\mathrm{in}} \sin(\theta_j) + \langle X_j \rangle_{\mathrm{in}} \cos(\theta_j) \tag{13}$$

The magnitude of this gradient is maximized when the initial state $|\psi_{\mathrm{in}}\rangle$ has a small $\langle Z_j \rangle_{\mathrm{in}}$ component and a large transversal component (e.g., $\langle X_j \rangle_{\mathrm{in}}$). This corresponds to states residing near the **equator** of the

---

[1] We use "principal-eigenvector projection" (loosely called "purification" in this paper) to denote $\bar{\rho} \mapsto \tilde{\rho} := |\phi_0\rangle\langle\phi_0|$ with $|\phi_0\rangle$ the top-eigenvector of $\bar{\rho}$. This is the closest pure state to $\bar{\rho}$ in fidelity.

Bloch sphere. An optimizer does not explicitly seek the equator, but the path of steepest descent is found in this region of high sensitivity. Therefore, preparing states that can be easily maneuvered towards the equator is a principled strategy for enhancing trainability.

### D.2 Geometric Interpretation of the Entanglement Strategy

We can interpret our data-driven strategy from the perspective of **Quantum Information Geometry**. The quantum encoding maps our classical data onto a **data manifold** $\mathcal{M}$ within the Hilbert space. The data-data entanglement layer, $U_M$, acts as a geometric transformation, or a "sculpting" process, that reshapes this manifold into a new one, $\mathcal{M}' = \{U_M|\psi(x)\rangle\}$.

**Role of Low HSD.** As established in *Proposition: Conditional benefit for low-HSD pairs* (see main text), selecting pairs with low HSD is equivalent to identifying dimensions of the data manifold that are already "aligned". Applying an entangling gate $U_M$ to these pairs acts as a **coordinated shear or twist** on the manifold. This transformation "prepares" the manifold $\mathcal{M}'$ such that a larger portion of it now lies in the high-sensitivity region (the equator, per Appendix D.1) for the subsequent ansatz.

**Role of High $\mathcal{I}_Q$.** The total correlation metric $\mathcal{I}_Q(i,j) = D_{\text{HS}}(\tilde{\rho}_{ij}, \tilde{\rho}_i \otimes \tilde{\rho}_j)^2$ measures the inherent geometric "deformation" of the data manifold in the $(i,j)$ subspace away from a simple product-state geometry. By choosing to entangle pairs with high $\mathcal{I}_Q$, we apply our sculpting tool ($U_M$) to regions of the manifold that are already rich in correlational structure, allowing the entanglement to amplify these existing non-trivial geometric features.

In summary, our data-driven structure search is a method for learning an optimal unitary transformation $U_{M^*}$ that performs **geometric engineering on the quantum data manifold**. The criteria of low HSD and high $\mathcal{I}_Q$ are principled guides for warping the manifold into a shape that is both more class-separable and geometrically poised for efficient optimization by the subsequent variational ansatz. *Note.* While higher correlation can make $U_M$ more potent, Appendix F shows that *overly* large correlation may suppress attainable parameter–QFI for common post-entangler generators. This motivates a tempered use of correlation in our utility.

## E  Appendix E. Low-HSD Entanglement and Trainability

We detailed the role of Hilbert–Schmidt distance (HSD) in enhancing the trainability of a QNN. We correct conjugation rules and refine the interpretation: the low-HSD criterion serves as a **geometric regularizer** that controls gradient Lipschitzness and mitigates barren plateau onset.

### Setup, Assumptions, and Notation

We consider a data-encoding circuit followed by an entangler $U_M$ and a parameterized local rotation. Let $\rho$ denote the pre-entanglement state and $\langle\cdot\rangle = (\cdot\,\rho)$.

**QFI convention.** For a pure input state and a unitary family with Hermitian generator $G$, we use $F_Q = 4\operatorname{Var}(G)$. For mixed inputs this becomes an upper bound, $F_Q \leq 4\operatorname{Var}(G)$; our lemmas below are equalities for pure inputs and read as upper bounds otherwise.

**Assumptions (E1–E4).**

(E1) **Single-axis expectation model.** For qubit $q$ there exist $(r_q, \phi_q)$ with $0 \leq r_q \leq 1$ such that $\langle X_q \rangle = r_q \sin\phi_q$ and $\langle Z_q \rangle = r_q \cos\phi_q$ (after a fixed local basis choice).

(E2) **Bounded-locality cost.** The measured observable is a $k$-local operator with $k = O(1)$.

(E3) **Moderate depth / light-cone locality.** The parameter generator $H$ has $O(1)$ light-cone.

(E4) **Weak inter-qubit factorization at initialization.** $\langle A_i \otimes B_j \rangle \approx \langle A_i \rangle \langle B_j \rangle$ for Pauli $A, B$ on $i \neq j$ (small covariances).

**Heisenberg conjugation for** $\mathrm{CX}_{ij}$ **(control** $i$**, target** $j$**).**

$$\mathrm{CX}_{ij}^\dagger X_i \, \mathrm{CX}_{ij} = X_i X_j, \quad \mathrm{CX}_{ij}^\dagger Z_i \, \mathrm{CX}_{ij} = Z_i, \tag{14}$$

$$\mathrm{CX}_{ij}^\dagger X_j \, \mathrm{CX}_{ij} = X_j, \quad \mathrm{CX}_{ij}^\dagger Z_j \, \mathrm{CX}_{ij} = Z_i Z_j. \tag{15}$$

(The analogous table for $\mathrm{CZ}_{ij}$ is $X_i \mapsto X_i Z_j$, $Y_i \mapsto Y_i Z_j$, $Z_i \mapsto Z_i$ and symmetrically for $j$.)

STEP 1: QFI GAIN FOR A PARAMETER ON THE CONTROL QUBIT

Let a post-entangler rotation on the control $i$ be generated by $X_i$; its effective generator on the pre-entangled state is $G'_i = U_M^\dagger X_i U_M = X_i X_j$. For pure inputs, $F_Q = 4 \, (G)$; for mixed inputs the following equalities become upper bounds.

**Lemma 1** (QFI after CX for control-$X$)**.**

$$F_{Q,after}^{(\theta_i^X)} = 4 \, (X_i X_j) = 4 \big( 1 - \langle X_i X_j \rangle^2 \big).$$

**Lemma 2** (QFI before CX for the same parameter)**.**

$$F_{Q,before}^{(\theta_i^X)} = 4 \, (X_i) = 4 \big( 1 - \langle X_i \rangle^2 \big) = 4 \big( 1 - r_i^2 \sin^2 \phi_i \big).$$

**Theorem 2** (Closed-form QFI gain for control-$X$ under (E1)–(E4))**.**

$$\Delta F_Q^{(i,X)} := F_{Q,after}^{(\theta_i^X)} - F_{Q,before}^{(\theta_i^X)} \approx 4 \, r_i^2 \sin^2 \phi_i \, \big( 1 - r_j^2 \sin^2 \phi_j \big),$$

*which reduces to* $4 \sin^2 \phi_i \, \cos^2 \phi_j$ *when* $r_i = r_j = 1$.

*Proof.* Using (E4) with $\langle X_i X_j \rangle \approx \langle X_i \rangle \langle X_j \rangle = r_i r_j \sin \phi_i \sin \phi_j$ and the lemmas: $4 \big( 1 - r_i^2 r_j^2 \sin^2 \phi_i \sin^2 \phi_j \big) - 4 \big( 1 - r_i^2 \sin^2 \phi_i \big) = 4 r_i^2 \sin^2 \phi_i \, \big( 1 - r_j^2 \sin^2 \phi_j \big)$. $\qquad\square$

**Target-$X$ parameter under** $\mathrm{CX}$ Because $U_M^\dagger X_j U_M = X_j$, the *QFI gain* for a target-$X$ parameter is $\Delta F_Q = 0$ (the QFI itself need not be zero).

STEP 2: LOCAL $\Delta F_Q$ VS. GLOBAL TRAINABILITY

Theorem 2 suggests that $\Delta F_Q$ is maximized when the control is near the equator and the target near a pole—i.e., a *high-HSD* pairing on the data manifold. While this locally boosts one parameter's Fisher information, repeatedly favoring such pairings across layers tends to induce increasingly nonlocal entanglement graphs with larger *effective* light-cones, which correlates with barren plateau behavior in deep or effectively global circuits McClean et al. (2018); Cerezo et al. (2021).

STEP 3: LOW-HSD AS A GEOMETRIC REGULARIZER (LIPSCHITZ GRADIENT BOUND)

Let $f(\theta; x) = \left(O\, U_\theta\, \rho(x)\, U_\theta^\dagger\right)$ be a $k$-local cost with $k = O(1)$ and $g(\theta; x) = \partial_\theta f(\theta; x)$ its gradient for a parameter with generator $H$. Define $\tilde{O}_\theta = U_\theta^\dagger O U_\theta$ and $\tilde{H} = U_\theta^\dagger H U_\theta$. For two *inputs* $x, x'$, write the Hilbert–Schmidt distance $D_{\mathrm{HS}}(\rho(x), \rho(x')) := \|\rho(x) - \rho(x')\|_2$.

**Proposition 1** (HSD–Lipschitz gradient (sample-to-sample)). *For any two inputs $x, x'$ and any parameter obeying (E2)–(E3),*

$$\left| g(\theta; x) - g(\theta; x') \right| = \left| \left(i[\tilde{H}, \tilde{O}_\theta]\, [\rho(x) - \rho(x')]\right) \right| \ \leq\ \|[\tilde{H}, \tilde{O}_\theta]\|_2\, D_{\mathrm{HS}}\!\left(\rho(x), \rho(x')\right),$$

*where locality bounds $\|[\tilde{H}, \tilde{O}_\theta]\|_2 \leq C$ for a constant $C$ independent of $n$.*

**Corollary 1** (Neighborhood entanglement preserves favorable scaling). *Suppose $U_M$ only entangles qubit pairs $(i, j)$ whose* local *reduced states remain HSD-close across the dataset, and the entanglement graph has maximum degree $\Delta = O(1)$. Then the dataset gradient field is $L$-Lipschitz (with $L = O(\Delta)$ times the typical local HSD scale), which helps* preserve *the known non-barren-plateau scaling for local costs and shallow effective depth Cerezo et al. (2021); Pesah et al. (2021).*

**Interpretation.**   Low-HSD restricts sample-to-sample $D_{\mathrm{HS}}$ within entangled neighborhoods, which (by Prop. 1) controls the Lipschitz constant of the gradient field. This trades potentially smaller *local* $\Delta F_Q$ for *global* optimization stability—precisely the regularization effect observed empirically.

STEP 4: A PRINCIPLED TRADE-OFF IN A UTILITY

We combine an informativeness term and a smoothness term:

$$\mathcal{U}_{\mathrm{data}}(E) = \sum_{(i,j)\in E} \underbrace{\mathcal{I}_Q(i,j)}_{\text{correlation/utility}} \ - \lambda \sum_{(i,j)\in E} \underbrace{D_{\mathrm{HS}}(\rho_i, \rho_j)}_{\text{geometric regularizer}} \ - \mu\ \underbrace{|E|}_{\text{gate budget}}\ , \tag{16}$$

with $\lambda, \mu > 0$. The first term prefers pairs where entanglement is potent; the second enforces manifold-locality (trainability); the third caps gate count or uses hardware-weighted cost. Greedy addition in decreasing $\mathcal{I}_Q - \lambda D_{\mathrm{HS}}$ naturally stops at the marginal-gain knee.

### E.1  SUMMARY

For CX, the gain $\Delta F_Q \approx 4 r_i^2 \sin^2 \phi_i \left(1 - r_j^2 \sin^2 \phi_j\right)$ is maximized by high-HSD pairing, but such pairing, when repeated, tends to nonlocal entanglement and barren plateau risks. Low-HSD acts as a *geometric regularizer*: it keeps the gradient field Lipschitz (Prop. 1) and preserves favorable scaling (Cor. 1), matching the empirical superiority of low-HSD / high-correlation search strategies.

## F  APPENDIX F. WHEN "TOO-HIGH" CORRELATION SUPPRESSES QFI

We formalize the intuition that overly large correlation (as measured by $\mathcal{I}_Q$) can *reduce* trainability proxies such as parameter–QFI for common entanglers.

**Setup.**   Let the pure-projected two-qubit state be $\tilde{\rho}_{ij}$ with Bloch marginals $r = \left(r_x, r_y, r_z\right)$ and $s = \left(s_x, s_y, s_z\right)$, and correlation tensor $T \in \mathbb{R}^{3\times 3}$ with $T_{ab} = \mathrm{Tr}(\tilde{\rho}_{ij}\, \sigma_a \otimes \sigma_b)$. Define the deviation from a product,

$$D := T - r s^\top \in \mathbb{R}^{3\times 3}.$$

In the projected (rank-1) setting, $\mathcal{I}_Q$ controls the Frobenius norm of $D$ via the Pauli expansion:

$$\boxed{\ \mathcal{I}_Q(i,j)\ =\ \tfrac{1}{4}\, \|D\|_F^2\ } \tag{17}$$

## F.1 EXISTENCE BOUND (WORST-CASE ALIGNMENT)

Let $M := \max_{a,b} |D_{ab}|$. By Cauchy–Schwarz, $M^2 \geq \frac{1}{9} \sum_{a,b} D_{ab}^2 = \frac{1}{9} \|D\|_F^2$, hence

$$M \geq \frac{1}{3} \|D\|_F = \frac{2}{3} \sqrt{\mathcal{I}_Q}. \tag{18}$$

Consider a controlled-phase entangler and a single-qubit parameter on $i$ whose post-entangler generator is $G = \sigma_a^{(i)} \sigma_b^{(j)}$ (e.g., CZ gives $X \mapsto XZ$). For a pure input, $F_Q = 4 \operatorname{Var}(G) = 4 \left(1 - \langle \sigma_a \otimes \sigma_b \rangle^2\right)$. Writing $T_{ab} = D_{ab} + r_a s_b$, we obtain the upper bound

$$F_Q \leq 4 \Big(1 - \big(|D_{ab}| - |r_a s_b|\big)^2\Big). \tag{19}$$

If the encoding is equator-centered for axis $a$ or $b$ (so that $|r_a s_b| \leq \varepsilon$), choose $(a,b)$ attaining $M$; then by equation 18–equation 19

$$F_Q \leq 4 \Big(1 - \big(\tfrac{2}{3} \sqrt{\mathcal{I}_Q} - \varepsilon\big)^2\Big), \tag{20}$$

exhibiting a *decrease* in the attainable QFI as $\mathcal{I}_Q$ grows. Thus there exist local axis choices (or equivalent pre-rotations) under which high $\mathcal{I}_Q$ suppresses QFI.

## F.2 AVERAGE-CASE BOUND (RANDOM LOCAL FRAMES)

Let $R, R' \in SO(3)$ be independent, Haar-uniform rotations corresponding to random local unitaries on $i$ and $j$. Under $(r, s, T) \mapsto (Rr, R's, RTR'^\top)$, we have $D \mapsto D' = RDR'^\top$ and Frobenius invariance $\|D'\|_F = \|D\|_F$. By symmetry of the $3 \times 3$ entries,

$$\mathbb{E}\big[D_{ab}'^2\big] = \frac{1}{9} \|D\|_F^2 = \frac{4}{9} \mathcal{I}_Q \qquad \text{for any fixed } (a,b). \tag{21}$$

For the post-entangler generator $G = \sigma_a \otimes \sigma_b$ we have $F_Q = 4(1 - T_{ab}'^2)$ with $T_{ab}' = D_{ab}' + r_a' s_b'$. Let $\delta := \mathbb{E}[(r_a' s_b')^2]$ (small for equator-centered encodings). Using Cauchy–Schwarz,

$$\mathbb{E}\big[T_{ab}'^2\big] = \mathbb{E}[D'^2] + \mathbb{E}[(r's')^2] + 2\,\mathbb{E}[D'r's'] \geq \big(\sqrt{\mathbb{E}[D'^2]} - \sqrt{\mathbb{E}[(r's')^2]}\big)^2 = \Big(\sqrt{\tfrac{4}{9} \mathcal{I}_Q} - \sqrt{\delta}\Big)^2.$$

Hence

$$\mathbb{E}[F_Q] \leq 4 \left(1 - \Big(\sqrt{\tfrac{4}{9} \mathcal{I}_Q} - \sqrt{\delta}\Big)^2\right) \leq 4 \Big(1 - \tfrac{4}{9} \mathcal{I}_Q + \delta\Big), \tag{22}$$

where the looser rightmost inequality follows from $(a - b)^2 \geq a^2 - b^2$. When isotropy renders the cross term mean zero, the middle expression simplifies directly to the looser bound with equality in the first step.

**Remarks.** (i) The suppression in equation 20 is an *existence* statement: if the dominant correlation component aligns with the generator's axes, QFI decreases with $\mathcal{I}_Q$. (ii) The average-case bound equation 22 formalizes a practical trade-off: high $\mathcal{I}_Q$ inflates typical correlators $|T_{ab}'|$, shrinking $F_Q$.

## G APPENDIX G. BI-LEVEL OPTIMIZATION ALGORITHM

**Problem setup.** Given dataset $\mathcal{D}$, hardware coupling graph $G_{\text{hw}} = (V, E_{\text{hw}})$, a target two-qubit budget $B$, and per-node degree hints $\tau = \{\tau_i\}_{i \in V}$, we seek an entanglement structure $M \subseteq E_{\text{hw}}$ and circuit parameters

$\theta$ that maximize a task objective $J(M, \theta)$ under $|M| \leq B$ and $\deg_M(i) \leq \tau_i$. We optimize a proxy score $\tilde{J}$ in the outer loop and use a Lagrangian with multipliers $\lambda_B \geq 0$ and $\{\lambda_i^{\mathrm{deg}} \geq 0\}$:

$$\mathcal{L}(M, \theta, \lambda) = J(M, \theta) - \lambda_B(|M| - B) - \sum_{i \in V} \lambda_i^{\mathrm{deg}}(\deg_M(i) - \tau_i).$$

Algorithm 1: Bi-Level Optimization for Entanglement Structure Search

**Require:** Dataset $\mathcal{D}$, Hardware graph $G_{\mathrm{hw}}$, Step sizes $\eta_B, \eta_{\mathrm{deg}}$, Outer rounds $T$
**Ensure:** Optimal entanglement structure $M_{\mathrm{best}}$

1: **Phase 1: Data-Driven Initialization**
2: Compute data utility $\mathcal{U}_{\mathrm{data}}(i, j)$ for all edges $(i, j) \in G_{\mathrm{hw}}$.
3: Compute node scores $s_i \leftarrow \sum_j \mathcal{U}_{\mathrm{data}}(i, j)$ for each qubit $i$.
4: Compute degree hints $\tau = \{\tau_i\}$ based on node scores $s_i$.
5: Set target budget $B \leftarrow \mathrm{round}(\frac{1}{2} \sum_i \tau_i)$.
6: Generate initial structure $M_0$ via greedy matching, respecting $\tau$ and $B$.
7: Initialize QNN parameters $\theta_0$.
8: Initialize Lagrange multipliers $\lambda_B \leftarrow 0$, and $\lambda_i^{\mathrm{deg}} \leftarrow 0$ for all $i$.
9: Initialize best solution trackers: $M_{\mathrm{best}} \leftarrow M_0$, $J_{\mathrm{best}} \leftarrow -\infty$.

10: **Phase 2: Bi-Level Optimization Loop**
11: **for** $t = 0, 1, \ldots, T - 1$ **do**
    *// — Inner Level: Full parameter training for current structure —*
12:    Freeze structure $M_t$. Train QNN parameters $\theta$ to find optimal $\theta_t^*$ for $M_t$.
13:    Evaluate the true objective value $J(M_t, \theta_t^*)$.
14:    **if** $J(M_t, \theta_t^*) > J_{\mathrm{best}}$ **then**
15:        $M_{\mathrm{best}} \leftarrow M_t$
16:        $J_{\mathrm{best}} \leftarrow J(M_t, \theta_t^*)$
17:    **end if**
    *// — Outer Level: Neighborhood search with proxy evaluation —*
18:    Generate a neighborhood of candidate structures $\mathcal{N}(M_t)$ around $M_t$.
19:    **for all** $M' \in \mathcal{N}(M_t)$ **do**
20:        Warm-start training from $\theta_t^*$ for a few epochs to get a proxy score $\tilde{J}(M')$.
21:    **end for**
22:    Select the best candidate from the neighborhood: $M_{t+1} \leftarrow \arg\max_{M' \in \mathcal{N}(M_t)} \tilde{J}(M')$.
    *// — Dual Update: Adjust multipliers based on constraint violations —*
23:    Calculate budget violation: $v_B \leftarrow |M_{t+1}| - B$.
24:    Calculate degree violations: $v_i^{\mathrm{deg}} \leftarrow \deg_o p_{M_{t+1}}(i) - \tau_i$ for all $i$.
25:    Update budget multiplier: $\lambda_B \leftarrow \max(0, \lambda_B + \eta_B \cdot v_B)$.
26:    Update degree multipliers: $\lambda_i^{\mathrm{deg}} \leftarrow \max(0, \lambda_i^{\mathrm{deg}} + \eta_{\mathrm{deg}} \cdot v_i^{\mathrm{deg}})$ for all $i$.
27:    **if** $M_{t+1} = M_t$ and all constraints are satisfied **then**
28:        **break**
29:    **end if**
30: **end for**
31: **return** $M_{\mathrm{best}}$

**Brief explanation.** The method alternates between (i) *inner-level* training of circuit parameters $\theta$ for a fixed entanglement structure $M$, and (ii) *outer-level* local search over structures guided by a proxy score $\tilde{J}$ that is evaluated via short warm-started fine-tuning. Budget and per-node degree constraints are enforced

softly through a Lagrangian: their multipliers ($\lambda_B$, $\lambda_i^{\mathrm{deg}}$) are updated by projected subgradient ascent on the corresponding violations and directly *shape* the adjusted edge weights $w'_{ij}$. This makes over-budget or over-degree edges less attractive in subsequent outer steps. We keep track of the best structure using the *full* objective $J$ (after inner training), while using $\tilde{J}$ only to cheaply rank local candidates.

# H  APPENDIX H. EXPERIMENTS SETTING

In this appendix, we explain more detail about our empirical experimental setting.

## H.1  SETUP

**Datasets.**  We evaluate all methods on two binary classification tasks:

- **Synthetic Dataset:** A non-linear, N-feature dataset with 1,000 samples, designed with complex feature correlations to specifically test the representation learning capabilities of the feature map. (Detailed below)

- **Real-World Dataset:** The UCI Heart Disease dataset with 13-features, a standard real-world benchmark, to assess performance on a practical problem.

For each dataset, we use a standard 70/30 train-test split and apply 'StandardScaler' to all features. All reported results are the mean and standard deviation over 4 independent runs with different random seeds.

**Encoding Methods.**  In our experiments, we benchmarked the performance of three distinct encoding methods were utilized:

- **Simple *RY* Encoding:** A fixed, manually designed circuit used as a baseline. This architecture encodes data using *RY* gates and applies entanglement via a linear chain of CNOT gates.

- **Data Re-uploading Structure:** Dense encoding methods that employs the data re-uploading principle, where data encoding layers and variational layers are strategically repeated.

- **Adaptive Pruned Structure:** A sparse circuit architecture which is obtained by starting with a more complex circuit and adaptively pruning gates based on a performance threshold.

**Entanglement Strategies.**  We compare the performance of the structure discovered by our framework (**Searched**) against several heuristic baselines:

- **Heuristic Baselines:** Linear chain structure and random selection are adopted to baselines.

- **Our Method (Searched):** This strategy is the output of our bi-level, multi-objective search algorithm, which is run independently for each dataset and hardware profile.

**Model and Training Details.**  The core model is an Hybrid QNN. The encoding layer maps the 8 input features using single-qubit 'RY' gates, followed by the selected entanglement structure. The variational ansatz consists of two layers, each containing single-qubit 'RX' and 'RZ' rotations on all qubits followed by a circular CNOT entanglement layer. For the final training of a selected architecture, all models are trained for 30 epochs using the Adam optimizer with a learning rate of $10^{-2}$ and a batch size of 32.

**Search Framework and Hyperparameters.**  Our bi-level search is configured as follows: the outer loop performs a discrete search over the space of 7-edge entanglement graphs. To evaluate each candidate structure, the inner loop conducts a proxy training for a brief 5 epochs. The final validation loss from this short training is used as the score to guide the outer loop. The hyperparameters for the multi-objective function ($\alpha, \beta, w_H$) are equally fixed as 1.0.

**Synthetic Dataset.**  To rigorously evaluate our method in a controlled setting with non-trivial dependencies, we procedurally construct a synthetic binary classification dataset. The dataset consists of $n_{\mathrm{samples}} = 1000$

examples and $n_{\text{total}} = 13$ features. The generation process is hierarchical, beginning with $n_{\text{base}} = 4$ independent base features sampled from a uniform distribution:

$$x_i \sim \mathcal{U}(-1, 1), \quad \text{for } i \in \{1, \ldots, n_{\text{base}}\}.$$

The remaining $n_{\text{total}} - n_{\text{base}}$ features are derived from this base set to introduce complex correlations. A specified fraction of these are **non-linear features**, generated by applying transformations such as products $(x_i \cdot x_j)$, trigonometric functions $(\sin(\pi x_i))$, and hyperbolic tangents $(\tanh(x_i - x_j))$ to randomly selected pairs of base features. The rest are **linear features**, formed from various weighted linear combinations of the base features. This process creates a feature matrix $\mathbf{X} \in \mathbb{R}^{n_{\text{samples}} \times n_{\text{total}}}$ with a rich, predefined correlation structure.

Finally, the binary labels $y \in \{0, 1\}$ are assigned based on a complex, non-linear decision boundary designed to be challenging for models that cannot capture higher-order feature interactions. The decision score $D(\mathbf{x})$ is computed as a weighted sum of several interaction terms across all features:

$$\begin{aligned}
D(\mathbf{x}) = {} & 0.7 \sin(4x_0 x_1) + 1.2(x_2^4 - x_3^3) \\
& - 0.9 \log(x_4^2 + x_5^2 + \varepsilon) \cdot \exp(x_6 + x_7) \\
& + 0.5 x_8 x_9 x_{10} - 0.6 \sqrt{|x_{11} x_{12}|}
\end{aligned}$$

where $\varepsilon$ is a small constant for numerical stability. The final label is assigned by thresholding the score against its median, ensuring a perfectly balanced dataset:

$$y = \mathbb{I}\left[ D(\mathbf{x}) > \text{median}\left( \{D(\mathbf{x}_i)\}_{i=1}^{n_{\text{samples}}} \right) \right].$$

# I  APPENDIX I. DETAILED ANALYSIS OF TRAINING DYNAMICS

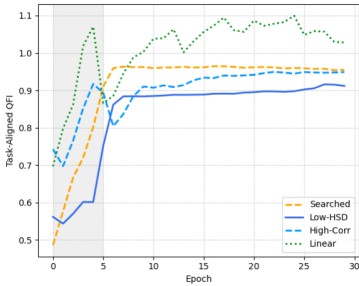

Figure 2: The task-aligned QFI, which measures the portion of QFI relevant to the learning task.

This section provides a detailed interpretation of the training dynamics for the Task-Aligned Quantum Fisher Information (QFI), as depicted in Figure 2. This metrics offer deeper insights into why our **Searched** strategy, along with other heuristics, achieves superior performance compared to the baseline.

The Task-Aligned QFI is a metric we compute to measure the portion of the model's expressive capacity that is effectively utilized for the specific learning task. Specifically, we calculate it using the following formula:

$$g(\theta) = \frac{(\nabla_\theta C)^T G(\theta)(\nabla_\theta C)}{||\nabla_\theta C||^2}$$

Here, $\nabla_\theta C$ is the gradient of the cost function $C$ with respect to the parameters $\theta$, and $G(\theta)$ is the Quantum Fisher Information matrix. This metric quantifies the magnitude of the QFI projected along the direction of the cost function's gradient. A higher value implies that the model's geometry is better aligned with the optimization objective.

In the plot, our **Searched** method demonstrates a clear advantage. After the initial epochs, it converges to the highest and most stable Task-Aligned QFI value. This confirms that the architectures discovered by our search algorithm are not just trainable, but are specifically optimized to direct their learning capacity towards solving the task at hand. The other heuristics, while also achieving reasonably high values, either plateau at a lower level (**Low-HSD**) or exhibit more volatility (**Linear**, **High-Corr**), indicating a less optimal alignment between the model's expressivity and the task.

