# OpenReview forum: "Data- and Hardware-Aware Entanglement Selection for Quantum Feature Maps in Hybrid Quantum Neural Networks"
_ICLR.cc/2026/Conference — ICLR 2026 Conference Withdrawn Submission_

### Official Review · Reviewer_KtZL · 2025-10-22

**Soundness:** 2
**Presentation:** 2
**Contribution:** 2
**Rating:** 2
**Confidence:** 4

**Summary:**

The authors present a framework for devising an entanglement strategy for a feature map of a QNN based on data and hardware.

The authors did not provide a background section introducing the setup of a QNN. I would expect the paper to at least describe the setup in terms of encoded features, ansatz, loss function and also to describe the BP phenomenon in more detail. Moreover, explanations for QFI and the content of the section on Theory-consistent design are missing. Further, there is a substantial amount of literature on quantum architecture search, QFI applications, hyperparameter optimization that were not referenced or discussed. The paper has structural issues. Large parts of the theory connecting important concepts of the main part are moved to the Appendix, making it hard to follow. I would urge the authors to refactor the paper. The appendix should be used to add additional, non-fundamental details, not to completely outsource large fractions of the paper.

Further, the benchmarking setup is lacking explanations. At least an introduction and overview of the experiments should be given in the main text (the appendix should only be used to give supplementary information, not to contain crucial aspects of the whole setup). The baselines do not represent what is commonly used (i.e., why use linear and random instead of common entanglement strategies that are implemented in Qiskit or PennyLane? pairwise, circular, etc. are what is commonly used while to the best of my knowledge random entanglement is never done). The results are missing statistical significance statements, in particular, considering that the differences between the methods are often only marginal.

The method is moreover not scalable, which is shortly explained in the conclusion by stating that for larger scale applications, direct state vector access is not a feasible alternative. Given that effort to retrieve and store the state vectors is exponential (as is full state tomography, which is noted in the conclusion as well), I am questioning why this is even proposed. I would be curious to hear how this method could be scaled to go even to twenty qubits. I think this is an important aspect that should be addressed in the rebuttal by the authors.

It is for these reasons that I cannot recommend the paper to be published at ICLR at this point.

**Strengths:**

- HW inspired architectures are indeed a promising research direction and could lead to significant benefits
- The objective function they propose provides a meaningful way to balance considerations in entanglement choices

**Weaknesses:**

- Abstract: "While entanglement in this feature map layer can enhance expressivity, heuristic choices often degrade trainability". Entanglement induced BPs are a proven phenomena - be more precise in what you want to state.
 - Please use \citep if you want to cite something inline, otherwise it is hard to read.
 - P.3: "Data-driven methods engineer the feature map to reflect data structure. This includes, ... , and data re-uploading". Data re-uploading does not engineer the feature map to necessarily reflect data structure, rather it just continuously encodes the input over and over.
 - Theory-consistent design: There are no further details on "manifold-local pairings" and, to me, it seems to be a claim that the HSD as a regularizer preserves favorable (non BP) scaling. This section needs further elaboration and background.
 - Line 181: referencing of Eq
 - Line 140: Please refrain from using "ideal" when used with heuristics
 - Elaborate on degree hints, it is not entirely clear what this means. Also, if this is a formula, why not just state it?
- No explanation on synthetic dataset generation in the main paper - a short description in the main text is at least necessary
- Exaggeration of results - the results do not demonstrate robustness of the approach if often times the results are only very marginally better than a random (!) baseline (Heart data AUC and Accuracy). Further, for the synthetic data, the main text states the performance compared to the worst baseline result which is misleading.
- Typos line 298

**Questions:**

- Line 129: Please elaborate on how utility and cost terms are normalized before being combined and why this is not shown in the equation.
- Line 130: Elaborate on the rank-based scaling that is employed.
- Eq 2: Define psi.
- Line 146: It does not suffice to state that the two-qubit DMs are computed in a similar process. State the process in the main part of the paper if it is such a significant part of the work. You can put details/derivations/further explanations in the Appendix but at least stating it in the main part would be appropriate.
- Line 166: What theoretical concerns are you mitigating? If stated like that, it requires at least a short explanation in the main paper before linking to the appendix.
- Line 169: The claimed connection between correlation and trainability seems to be only on empirical observations. Is there any theory to back this up? Otherwise, it needs to be framed differently to mirror that this is only an empirical observation.
- Eq 6: Why is the accumulated error from the SWAP sequences 3? Can you elaborate in the paper?
- Line 209: Why is the full objective function not restated? This seems to be an important aspect.
- Why is random entanglement not integrated into ATP and Data Reuploading?
- How is random entanglement enforced? What is the strategy? How often do you repeat these results to get any sort of robustness?
- With randomization involved, can you elaborate on the statistical significance?
- What is the Silhouette score in Fig1a and why is not explained before?
- Why are the training dynamics in 4.2 based on different experiments than 4.1? Again, missing details on experimental setup.
- Line 321: Why would better convergence be evidenced by a higher class separability? How are the two connected?
- Line 325: Why is the central hypothesis only detailed in the Appendix? This is an important aspect of the work, while the Appendix is only supposed to provide additional information.
- Line 326: Elaborate on connection of QFI with optimization direction?
- Line 332: What does "our proposed method exhibits rapidly increase in trajectory" mean? Faster convergence? Please clarify
- Section 4.3.: I am missing any explanation of the ablation study
- Table 2: The differences in performance seem very marginal. Please elaborate on the statistical significance.
- Line 408: How would IQ and HSD criteria ensure global trainability. Please be more precise.
- Line 411: The theory and experiments do not confirm that the architecture is fundamentally trainable and the experiments show that it is often only marginally better than random entanglement. Therefore, it is unjustified to claim superior performance - in particular, since also the experiments are very limited.
- Line 417: please elaborate on the effects of realistic noise. You only consider two-qubit errors, which is not enough to state that it will work welll on NISQ.
- Line 422: How would you employ state tomography? This most generally involves constructing the full DM, so I do not understand why it would be a vital and practical avenue?

---

### Official Review · Reviewer_Cb3s · 2025-10-24

**Soundness:** 2
**Presentation:** 2
**Contribution:** 2
**Rating:** 2
**Confidence:** 4

**Summary:**

This paper proposes a framework for optimizing entanglement structures in the data-encoding layer of Hybrid Quantum Neural Networks (HQNNs). The key idea is to formulate entanglement selection as a multi-objective optimization problem balancing data-driven trainability, hardware noise robustness, and circuit efficiency. The approach introduces a data utility term based on Hilbert-Schmidt Distance (HSD) and an intrinsic dependency metric (IQ), combined with a hardware-aware cost computed from calibrated IBM Quantum backend fidelities. A bi-level optimization process searches discrete entanglement structures, guided by short inner-loop training runs. Experiments on synthetic and real datasets show improved accuracy and robustness compared to baseline entanglement patterns.

**Strengths:**

1.	Addresses a practically important problem for near-term quantum hardware by integrating data- and hardware-aware considerations.
2.	The proposed multi-objective formulation is conceptually coherent and unifies several recent heuristics under one framework.
3.	The experimental section includes multiple encoding strategies (Angle, ATP, and Data Re-uploading), showing generality of the application.

**Weaknesses:**

1.	In Section 3.4, the algorithm’s convergence behavior is not discussed. There is no theoretical or empirical evidence that the alternating updates between outer and inner levels lead to consistent improvement or stable optima. The paper should add convergence diagnostics or comparison with reinforcement-based search.
2.	The use of rank-based scaling across objectives in Equation (1) is insufficiently justified. It may introduce bias depending on dataset size or search granularity. A sensitivity analysis to α, β parameters is missing.
3.	Figure 1 lacks numerical labels and units for the Silhouette Score, Gradient Norm, and Tr(QFI). The interpretation in the text (“robust quantum gradient norm”) remains qualitative. Quantitative evidence (e.g., rate of gradient decay) would strengthen claims.
4.	In the Ablation Study in Table 2, the claim that combining IQ and HSD “synergistically” improves trainability is not strongly supported. The difference between “Full Method” and “HSD Only” is marginal.
5.	The noise model is limited to 2-qubit gate errors and neglects readout and single-qubit noise, which dominate in IBM hardware. Consequently, the conclusion of “hardware robustness” is not appropriate.
6.	Many mathematical derivations are restatements of standard QFI-gradient relationships without formal proofs or novel insights. The theoretical justification does not clearly support the empirical advantage claimed in Section 4.2.
7.	The limitation that the computational complexity of computing pairwise HSD and IQ metrics scales as O(n²) in qubits is only mentioned but not quantified.
8.	All experiments use simulators. No experiments using a real device are shown despite claiming “hardware-aware optimization.” A small-scale run on IBM Q hardware would greatly increase credibility.
9.	The discussion of “future scalability” is generic. Concrete numerical estimates or runtime comparisons would help evaluate feasibility on larger qubit systems.

**Questions:**

1.	How sensitive is the bi-level optimization to hyperparameters α, β and the number of outer iterations?
2.	How does the proposed method compare to differentiable quantum architecture search or reinforcement learning-based approaches?
3.	Could the runtime complexity or search-space statistics be provided to quantify computational cost relative to brute-force search?
4.	Would incorporating realistic noise calibration (including readout errors) change the conclusions in Table 3?
5.	Can the framework be extended to hybrid ansätze beyond angle encoding?

---

### Official Review · Reviewer_BzRq · 2025-11-01

**Soundness:** 2
**Presentation:** 1
**Contribution:** 2
**Rating:** 2
**Confidence:** 3

**Summary:**

This paper presents a data- and hardware-aware framework for selecting entanglement structures in hybrid quantum neural networks (HQNNs). It formulates entanglement selection as a multi-objective optimization problem that balances data-driven trainability, hardware cost, and circuit efficiency. By integrating the Hilbert–Schmidt Distance (HSD) and a quantum correlation metric (IQ), the proposed bi-level search algorithm automatically identifies entanglement patterns that improve performance, robustness, and efficiency on realistic quantum devices. Experiments on synthetic and real datasets demonstrate significant gains in accuracy, trainability, and noise resilience compared to heuristic baselines.

**Strengths:**

Na

**Weaknesses:**

1. The manuscript suffers from severe issues in clarity and technical presentation. Many critical terms and symbols are undefined or inconsistently introduced, making it difficult to follow the proposed framework and assess its validity. Specifically:

- The terms $\mathcal{U}_{data}$, $C_{hardware}$, $R_{eff}$ in Eq. (1) are not properly defined, leaving their mathematical or physical meanings unclear.

- The notation $\mathcal{D}$ in Eq. (2) is introduced without explanation.

- The weights $w_{corr}$ and $w_H$ in Eq. (5) are presented without specifying their definition.

- While the authors denote $M$ as the structure of QNNs, the specific mathematical description of $M$ is not given. How to understand the notations $(i,j) \in M$?

- The parameter $N_{CNOT}^{SWAP}$ in Eq. (6) appears without any accompanying explanation of its computation or physical significance.

Overall, the presentation quality is poor, and the manuscript appears to be insufficiently prepared for formal submission. The lack of rigorous definitions, consistent notation, and clear exposition severely undermines the readability and credibility of the work.

2. While the paper introduces a seemingly principled multi-objective optimization framework, its theoretical underpinnings remain vague and insufficiently justified. Key formulations, such as the bi-level optimization structure and the data-utility function, are presented without clear derivations, assumptions, or proofs of convergence. The connection between the Hilbert–Schmidt Distance (HSD) and Quantum Fisher Information (QFI) is asserted but not rigorously demonstrated within the manuscript.

3. The experimental evaluation is weak and lacks sufficient evidence to support the claimed contributions. All results are obtained on small-scale simulators with limited datasets (e.g., synthetic and UCI Heart), which fail to demonstrate scalability or generalization to more realistic quantum or hybrid settings.

**Questions:**

The questions are included in the weakness.

---

### Official Review · Reviewer_CVYV · 2025-11-08

**Soundness:** 2
**Presentation:** 2
**Contribution:** 2
**Rating:** 4
**Confidence:** 4

**Summary:**

This paper presents an approach to quantum kernel methods through Quantum Generator Kernels (QGKs), leveraging variational generator groups for parameter-efficient quantum embeddings. The work demonstrates theoretical foundations with rigorous Lie-algebraic framework and comprehensive experimental validation across multiple datasets, showing superior performance over classical and quantum baselines. The concept of generator-based kernels addresses a critical scalability challenge in quantum machine learning.

**Strengths:**

This paper presents a highly innovative approach to quantum kernel methods through Quantum Generator Kernels (QGKs), leveraging variational generator groups for parameter-efficient quantum embeddings. The work demonstrates strong theoretical foundations with rigorous Lie-algebraic framework and comprehensive experimental validation across multiple datasets, showing superior performance over classical and quantum baselines. The concept of generator-based kernels addresses a critical scalability challenge in quantum machine learning.

**Weaknesses:**

1.	NISQ Hardware Practicality: The compiled circuit depths (e.g., 4,455 gates for 5-qubit MNIST task in Table 3) significantly exceed current NISQ device capabilities. While the hybrid compression strategy is proposed, actual hardware validation or concrete depth-reduction techniques are lacking, undermining near-term applicability claims.
2.	Baseline Comparison Fairness: The HEE baseline encodes only 5 features versus QGK's 93 for MNIST, creating an unfair comparison due to vastly different classical preprocessing burdens. This skews performance comparisons and should be addressed through balanced feature encoding or explicit discussion of this limitation.
3.	Noise Robustness Gap: The theoretical framework assumes perfect generator commutativity, but no analysis is provided on how realistic hardware noise might disrupt this property or affect kernel performance. Noise simulation results remain limited to ideal conditions.

**Questions:**

1.	NISQ Hardware Practicality: The compiled circuit depths (e.g., 4,455 gates for 5-qubit MNIST task in Table 3) significantly exceed current NISQ device capabilities. While the hybrid compression strategy is proposed, actual hardware validation or concrete depth-reduction techniques are lacking, undermining near-term applicability claims.
2.	Baseline Comparison Fairness: The HEE baseline encodes only 5 features versus QGK's 93 for MNIST, creating an unfair comparison due to vastly different classical preprocessing burdens. This skews performance comparisons and should be addressed through balanced feature encoding or explicit discussion of this limitation.
3.	Noise Robustness Gap: The theoretical framework assumes perfect generator commutativity, but no analysis is provided on how realistic hardware noise might disrupt this property or affect kernel performance. Noise simulation results remain limited to ideal conditions.

---

### Note · Authors · 2025-11-18

I have read and agree with the venue's withdrawal policy on behalf of myself and my co-authors.